# Synthesis, Characterization, and In Vivo Study of Some Novel 3,4,5-Trimethoxybenzylidene-hydrazinecarbothioamides and Thiadiazoles as Anti-Apoptotic Caspase-3 Inhibitors

**DOI:** 10.3390/molecules27072266

**Published:** 2022-03-31

**Authors:** Sara M. Mostafa, Ashraf A. Aly, Stefan Bräse, Martin Nieger, Sara Mohamed Naguib Abdelhafez, Walaa Yehia Abdelzaher, El-Shimaa M. N. Abdelhafez

**Affiliations:** 1Chemistry Department, Faculty of Science, Minia University, El-Minia 61519, Egypt; sara.ahmed@mu.edu.eg; 2Institute of Organic Chemistry, Karlsruhe Institute of Technology, 76131 Karlsruhe, Germany; 3Institute of Biological and Chemical Systems (IBCS-FMS), Karlsruhe Institute of Technology, Eggenstein-Leopoldshafen, 76131 Karlsruhe, Germany; 4Department of Chemistry, University of Helsinki, P.O. Box 55 (A. I. Virtasen aukio I), 00014 Helsinki, Finland; martin.nieger@helsinki.fi; 5Department of Histology, Faculty of Medicine, Minia University, El-Minia 61519, Egypt; sara_histology@yahoo.com; 6Department of Pharmacology, Faculty of Medicine, Minia University, El-Minia 61519, Egypt; walaayehia22@yahoo.com; 7Department of Medicinal Chemistry, Faculty of Pharmacy, Minia University, El-Minia 61519, Egypt; shimaanaguib_80@yahoo.com

**Keywords:** hydrazinecarbothioamides, thiadiazoles, antiapoptotic activity, caspase-3, selectivity, histochemical staining, molecular docking

## Abstract

The present study aims to discover novel derivatives as antiapoptotic agents and their protective effects against renal ischemia/reperfusion. Therefore, a series of new thiadiazole analogues **2a**–**g** was designed and synthesized through cyclization of the corresponding opened hydrazinecarbothioamides **1a**–**g,** followed by confirmation of the structure via spectroscopic tools (NMR, IR and mass spectra) and elemental analyses. The antiapoptotic activity showed alongside decreasing of tissue damage induced by I/R in the kidneys of rats using *N*-acetylcysteine (NAC) as an antiapoptotic reference. Most of the cyclized thiadiazoles are better antiapoptotic agents than their corresponding opened precursors. Particularly, compounds **2c** and **2g** were the most active antiapoptotic compounds with significant biomarkers. A preliminary mechanistic study was performed through caspase-3 inhibition. Compound **2c** was selected along with its corresponding opened precursor **1c**. An assay of cytochrome *C* revealed that there is an attenuation of cytochrome *C* level of about 5.5-fold, which was better than **1c** with a level of 4.1-fold. In caspases-3, 8 and 9 assays, compound **2c** showed more potency and selectivity toward caspase-3 and 9 compared with **1c**. The renal histopathological investigation indicated normal renal tissue for most of the compounds, especially **2c** and **2g,** relative to the control. Finally, a molecular docking study was conducted at the caspase-3 active site to suggest possible binding modes.

## 1. Introduction

Thiocarbohydrazides and their derivatives appear to be ideal candidates for the synthesis of new analogues and biological activity investigation, since they are the core feature in families of compounds known to display biological activities [1,2,3,4,5,6,7,8]. In recent years, thiocarbohydrazide derivatives have been versatile and convenient precursors for the synthesis of heterocyclic compounds, such as five membered rings [9,10,11,12], six membered rings [13,14,15], seven membered rings [16,17], and fused heterocycles [18,19] as well as for the synthesis of transition metal complexes [4,20] and pharmacological studies [21]. On the other hand, nitrogen-containing heterocycles are indispensable structural units for medicinal chemists. The chemical and biological applications of 1,3,4-thiadiazole has been subjected to intense studies from different research groups. Substituted 1,3,4-thiadiazoles have attracted significant interest in medicinal chemistry due to their wide range of pharmaceutical and biological activities, including antihypertensive [22], anti-inflammatory [23], antimicrobial [24], antitubercular [25], antifungal [26], anticonvulsant [27], anticancer [28], anti-HIV [29], antileishmanial [30], antidepressant [31], and antipsychotic [32] activities. 1,3,4-Thiadiazole derivatives also exhibit interesting in vivo and in vitro antitumor activities [33,34]. The synthesis of 1,3,4-thiadiazoles usually involves multi-step procedures, such as the cyclization of thiosemicarbazide with dicyclohexyl-carbodimide (DCC) and di-(2-pyridyl)thionocarbonate (DPT) [35]. 1,3,4-Thiadiazole derivatives are also formed by oxidative cyclization of thiosemicarbazides with FeCl_3_ [36] or by the reaction of thiosemicarbazides and CS_2_. The treatment of isothiocyanates with lithiated (trimethylsilyl)-diazomethane (Me_3_SiCN_2_Li) in Et_2_O, 2-amino substituted-1,3,4-thiadiazoles were obtained [37,38]. 

For several decades apoptosis has been known to be an important vital biological process that occurs in living cells during tissue development, immune responses [39,40], and homeostasis [41]. Anti-apoptotic mediators can be abrogated in pathological states, as in cancer [42] and autoimmunity [43], or exacerbated, as in stroke [44], neurodegeneration [45], retinal cell death [46], rheumatoid arthritis [47], myocardial and liver ischemia [48,49], and inflammatory diseases such as sepsis [50], osteoarthritis (OA) [51], and asthma [52]. 

The caspases family is a group of apoptotic mediators, which for many years have been considered high-priority targets for designing new target compounds. The dysregulation of caspase activity leads to many severe diseases. We focused on studying caspase-3, 8, and 9, as they play a key executioner role, and their inhibition can drastically prevent apoptosis in vitro and in vivo [53].

Renal ischemia-reperfusion usually occurs due to trauma, sepsis, renal transplantation, and some vascular surgeries, and this situation may be a reason for renal failure [54]. Although various mechanisms have been reported about the pathogenesis of I/R injury, the data about the treatment are very limited. The ischemia of the kidney starts a series of incidents, including cellular dysfunction and necrosis [55]. The combination of apoptosis and necrosis results in the death of renal cells, and ischemia-reperfusion injury (IRI) is the major cause of delayed graft function in renal allografts [56] in addition to histological damages and functional disorders due to regaining blood flow after an ischemic period in the kidney [57]. 

Several publications have appeared in the recent years documenting that variant of heterocycles play an important role in protecting complications accompanied with renal ischemia/reperfusion. Thiadiazolo-pyrazole **I** exerts a protective action against ischemia via the inhibition of NF-κB that enhances the process of cell apoptosis [58] (Figure 1). In addition, other heterocycles such as triazolopyrimidine derivative **II** (Figure 1) were tested in a kidney ischemia-reperfusion model, in which it showed efficacy at a dose of 10 mg/kg. Moreover, compound **II** is protective in a model of renal fibrosis [59]. Furthermore, it was reported that melatonin **III** treatment on renal I/R injury had putative protective effects, whereas melatonin administration reversed the oxidant responses and improved renal function and microscopic damage. It seems likely that melatonin protects kidney tissue against oxidative damage [60] (Figure 1).

Because of the emergence of acute renal ischemia and its ongoing impact on the global health system, there has been a growing interest for the development of novel antiapoptotic agents in order to combat this disorder. Thus, we report here our medicinal chemistry efforts, starting with the core scaffold of the caspase inhibitor that led to the design and synthesis of new thiadazole analogues as simpler and more efficient antiapoptotic compounds (Figure 2). Moreover, the aim of this study is to test the activity of the cyclized thiadiazole derivatives **2a**–**g** in comparison to their corresponding opened hydrazine-carbothioamides **1a**–**g** in terms of screening antiapoptotic activity, molecular modelling studies, and caspase-3, 8, and 9 inhibition assays to study the plausible mechanism of action of the newly synthesized compounds (Figure 2).

## 2. Results and Discussion

### 2.1. Chemistry

When equimolar amounts of 3,4,5-trimethoxybenzylidenethiocarbonohydrazide (**3**) [61] and 2-substituted isothiocyanates **4a**–**g** were stirred in dry DMF at room temperature, compounds (*E*)-*N*-(substituted)-2-(2-(3,4,5-trimethoxybenzylidene)hydrazinecarbonothi- oyl)hydrazinecarbothioamides **1a**–**g** were obtained in good yields (Figure 1).

The structure of the obtained products was elucidated by NMR, IR, and mass spectra, in addition to elemental analysis. For example, the ^1^H NMR spectrum of **1c** displayed five different broad signals with the integral ratio 1:1:1:1:1 centered at *δ_H_* = 11.87, 10.13, 9.43, 7.97, and 7.90 ppm (exchangeable with D_2_O) due to ^4^NH, ^3^NH, ^6^NH, cyclopropyl-NH, and CH=N, respectively. The ^1^H NMR spectrum of **1c** clearly indicated the presence of the cyclopropyl group, which appeared as two multiplets resonated at *δ_H_* 0.60–0.65 and 3.02–3.06 ppm due to 2CH_2_ and CH, respectively. The three methoxy groups appeared at *δ_H_* = 3.69 and 3.83 ppm as two methoxy proton sets, which were overlapped. The presence of the cyclopropyl group was also evident from the ^13^C-DEPT NMR spectrum exhibiting a positive signal at *δ_H_* = 27.18 ppm (CH) and negative signals at *δ_H_* = 6.28 ppm due to (2CH_2_). The formation of **1a**–**g** was evident by both the presence of a signals at *δ_C_* = 182.54 and 179.26 ppm attributed to the C=S group in the ^13^C NMR spectrum. Distinctive ^13^C NMR carbon signals of **1c** were appeared at *δ_C_* = 153.07 (Ar-C-O), 142.93 ppm for (CH=N), 141.84, 139.10, 129.35 (Ar-C), 104.87 (Ar-CH), 56.08, and 60.05 ppm (OCH_3_).

Synthesis of 5-(2-(3,4,5-trimethoxybenzylidene)hydrazinyl)-1,3,4-thiadiazol-2-amines **2a**–**g**, was established via the heterocyclization of (*E*)-*N*-substituted-2-(2-(3,4,5-trimethoxybenzylidene)hydrazinecarbonothioyl)hydrazinecarbothioamide **1a**–**g** during gentle heating in absolute ethanol. Accordingly, the assigned compounds **2a**–**g** were obtained as the only obtained products in 89–94% yields (Figure 1). The isolated compounds **2a**–**g** (Figure 1) were characterized by their IR, ^1^H NMR, ^13^C NMR, and mass spectral data in addition to elemental analysis. As for example, the IR spectrum of **2c** was characterized by the presence of a broad NH at *ν* = 3305-3250, and a band at *ν* = 1618 cm^−1^ that attributed to the C=N stretching. The bands attributed to C=S stretching vibrations were not observed in the IR spectra of **2a**–**g**. The chemical shifts obtained from the ^1^H NMR spectrum of **2c** support the proposed structure. Resonance attributed to a propyl group was detected at *δ_H_* = 0.73–0.88 (2 CH_2_-cyclopropyl) and 3.07–3.08 ppm (CH-cyclopropyl). The ^1^H NMR spectrum clearly showed a singlet signal at *δ_H_* = 7.96 due to CH=N, whereas two single broad bands with exchangeable D_2_O were observed at *δ_H_* = 8.32 and 11.56 ppm for the NH-propyl and NH-hydrazine, respectively. ^13^C NMR spectroscopic data of representative compounds **2a**–**g**, which were obtained using the DEPT technique at 100 MHz, also supported the carbon framework by discrimination of CH_2_, CH, and quaternary carbons. The ^13^C NMR spectrum of **2c** showed downfield signals at *δ_C_* = 166.64, 160.05, 153.07, 142.04, 104.77, and 26.95 ppm attributed to thiadiazole-C2, thiadiazole-C5, Ar-C-O, CH=N, Ar-CH, and CH-cyclo-propyl, respectively. The upfield signal resonated at *δ_H_* = 6.78 ppm due to (CH_2_-cyclopropyl).

The *E*-form of compounds **2a**–**g** was proved from X-ray structure analysis of (*E*)-*N*-phenyl-5-(2-(3,4,5-trimethoxybenzylidene)hydrazinyl)-1,3,4-thiadiazol-2-amine (**2d**) (Figure 3). Tables of X-rays are shown in Appendix A.

### 2.2. Evaluation of Biological Activity

This part describes the antiapoptotic effect of the novel target compounds **2a**–**g** in comparison to their intermediates **1a**–**g** on decreasing tissue damage induced by I/R in kidneys of rats in a dose equivalent to 30 mg *N*-acetylcysteine (NAC) [62] as an antiapoptotic reference [63]. Recently, it was reported that NAC was used as an antiapoptotic reference in estimating the apoptotic inhibition of new quinolone-based heterocycles, such as pyrazole and triazole [64]. The rats were subdivided into sham, I/R, NAC, and treated with tested compounds groups, and each group comprises six rats (weighing 140–150 g). The tested compounds were administrated intra-peritoneal (i.p) an hour before ischemia (0.5 h) and reperfusion (0.5 h), after which the organs were cut (Figure 4). This study is supported by biochemical, histological, and morphometric studies using different types of biomarkers.

Some of these biomarkers increased, and others decreased, indicating antiapoptotic effects, as illustrated in Table 1.

Appendix A indicate that **I/R**, **1b**, **1g**, **2b** and **2e** groups displayed significant increases in serum creatinine and urea and significant decreases in EP when compared to control and NAC groups. Meanwhile **1a**, **1c**, **1d**, **1e,** and **1f** groups displayed significant decreases in creatinine and urea and significant increases in EP when compared to the I/R group. The same table lists that groups **I/R**, **1b**, **1c**, **1f**, **1g**, **2b**, **2c**, **2e**, and **2f** had a significant increase in MDA along with a significant decrease in NOx and TAC when compared to control and NAC groups. Groups **1a**, **1d**, and **1e** showed significant improvements in oxidative stress parameters when compared to the I/R group. On the other hand, groups **1c** and **1f** showed significant results in MDA, NOx and TAC when compared to I/R group.

#### 2.2.1. Inhibition Effect on Serum Creatinine, Urea, and Renal MDA

The present study was performed to investigate the validity of serum creatinine levels as an indicator of postischemic renal dysfunction in mice [65]. The results show that the cyclized compounds **2c** and **2g** had excellent serum creatinine, serum urea, and renal MDA inhibition effects with conc 1.05 and 2.1 mg/dL (creatinine); 37.41 and 57.28 mg/dL (urea); and 54.55 and 84.01 nmol/gm (tissue) (renal MDA), which were respectively better than their corresponding opened intermediates **1a** and **1g** of conc 1.8 and 3.41 mg/dL (creatinine); 58.87 and 95.69 mg/dL (urea); and 85.48 and 154 nmol/gm (tissue) (renal MDA) and better than those, respectively, in comparison to NAC (3.18 mg/dL for creatinine, 92.17 mg/dL for urea, and 153.1 for renal MDA) (Figure 5).

Compounds **2e** and **2a** showed moderate inhibition to creatinine and urea when compared to their corresponding opened intermediates **1e** and **1a**, respectively. Compounds **2b**, **2d**, and **2f** exhibited fewer inhibition effects to all biomarkers (creatinine, urea, and MDA) than their corresponding opened intermediates **1b**, **1d**, and **1f** (Figure 5 and Appendix A).

#### 2.2.2. Activation Effect on Serum Epinephrine, Renal NOx, and Serum TAC

The assay of activation parameters (serum epinephrine, renal NOx, and serum TAC) was conducted for the treated groups of opened **1a**–**g** and cyclized compounds **2a**–**g** in comparison with sham and NAC, and the results are outlined in Appendix A. Both cyclized compounds **2c** and **2g** had excellent epinephrine activation effects with 44.51 and 27.39 ng/mg protein, respectively, in comparison with their corresponding opened intermediates **1a** and **1g** of conc 26.58 and 14.05 ng/mg protein, as illustrated in Figure 6 and Appendix A.

Moreover, compounds **2c** and **2g** had excellent renal NOx and serum TAC activation effect with conc 133.2 and 85.57 nmol/gm tissue (NOx) and 3.14 and 1.56 mmol/L (TAC), respectively, relative to their corresponding opened intermediates **1a** and **1g** of conc 81.78 and 56.37 nmol/gm tissue (NOx). Meanwhile, these activation biomarkers were moderately increased in the model of **2b** when compared to the opened form **1a**; however, the cyclized forms of **2d** and **2f** showed lesser activation in comparison with the opened forms of **1d** and **1f** treated groups, as illustrated in Appendix A.

#### 2.2.3. Caspases-3, 8, and 9 Inhibition and Selectivity

The above findings indicate remarkable antiapoptotic activity of our target compounds and suggest that the inhibition of apoptotic caspases may be the suggested target mechanism. Caspases can delay the apoptosis condition and thus implicating a potential role in drug screening efforts [20]. Cell-based caspase-3, and proapoptotic caspase-8 and 9 can cleave the synthetic substrate to release free AFC (7-amino-4-(trifluoromethyl)coumarin) as well as selectivity of caspase-3 and cell-permeable fluorescent reporters can be quantified by fluorometry. The compounds that are to be screened can directly be added to the reaction, and the level of inhibition of caspase-3 activity can be determined by a comparison of the fluorescence intensity in samples with and without the testing inhibitors. Therefore, the mechanism of the caspase-3, 8, and 9 inhibition activity and selectivity in MOLT-4 Cell Line was investigated. 

To outline the mechanism of the most active antiapoptotic cyclized thiadiazol derivative **2c** and its relative opened hydrazinecarbothioamides **1c**, whether through the inhibition of the intrinsic or the extrinsic pathways or both, their effects on caspase-8 and caspase-9 were also evaluated in comparison with the control and the NAC-treated group, as shown in Appendix A. Moreover, to investigate caspase-3 inhibition as well as selectivity being the suggested mechanism, the cyclized thiadiazol derivative **2c** and its relative opened hydrazinecarbothioamide **1c** were further studied for their effects on caspase-3, 8, and 9 and compared to the reference NAC.

Results indicate that compound **2c** showed downregulation in the level of active caspase-3 and caspase-9 with conc = 69.77 and 17.58 ng/mL (fold change = 3.1, 3.41), which was better than **1c** with conc = 163.1 and 50.6 ng/mL (fold change = 1.33, 1.19), respectively, and better when compared to NAC (conc = 135.2, 46.83 ng/mL, respectively), as shown in (Appendix A, Appendix A). 

Moreover, the effects of compound **2c** and **1c** on caspases-8 were also evaluated, revealing a slight increase in the levels of caspases-8 with conc = 35.57 and 20.56 ng/mL, relative to those of NAC, respectively, and compared to those of the untreated control (Figure 7) and (Appendix A, Appendix A).

#### 2.2.4. Assay of Cytochrome *C* Inactivation

For more proof of caspase-3 inhibition by the synthesized compounds, cell-based cytochrome *C* was assayed in an SR human cell line. It was reported that cytochrome *C* concentrations in the cell have a critical role in the inactivation of caspases and prohibiting the intrinsic apoptosis pathway [67]. Antiapoptotic proteins can inhibit apoptosis by blocking the release of cytochrome *C*, whereas proapoptotic members function as activators of its release. When cytochrome *C* levels are maintained in mitochondria the corresponding caspase-8 and 9 will not be activated that leads to the inactivation of caspase-3 and ceases the apoptosis process and therefore reserves the cell from programmed death [68].

The thiadiazole derivative **2c** and its opened precursor **1c** were evaluated for cytochrome *C* against the L-SR human cell line, and the results are listed in Figure 8. Compound **2c** caused a down-expression of cytochrome *C* levels about 5.5-fold better than **1c** (4.1-fold), compared to NAC (6.19-fold), and lower than the control. The displayed results can be a good guide for suggesting that antiapoptotic activity may be attributed to the attenuation of cytochrome *C* and hence inactivation of the intrinsic apoptotic pathway induced by the tested compounds (Figure 8).

##### Structure Activity Relationship

It can be concluded from the previous results that cyclized thiadiazoles heterocycles showed antiapoptotic activity better than their corresponding opened hydrazinecarbothioamides. It is noted that thiadiazoles of 3–5 carbon chains, such as **2c** and **2g** which were substituted with saturated alicyclic (cyclopropyl) or aliphatic groups (pentyl), showed the highest activity, but there was activity to a lesser extent when substituted with carbons chain <3, carbons as an ethyl group (compound **2b**), cyclohexyl (compound **2e**), or an aromatic benzyl group (compound **2a**) (Figure 9).

#### 2.2.5. Histopathological Investigation

From the previous work, consistent results motivated us to carry out a histopathological investigation of the tested compounds in the kidney. The groups numbers described below are represented as the following: gp1 = control, gp2 = I/R, gp3 = NAC, gp4 = 2a, gp5 = 2b, gp6 = 2c, gp7 = 2d, gp8 = 2e, gp9 = 2f, and gp10 = 2g.

##### Renal Cortex

The Sham group showed normal organization of the renal cortex. The glomerular capillary tufts, the parietal layer of Bowman’s capsule, and the Bowman’s space were studied. The proximal convoluted tubules (PCTs) had narrow lumen and a highly acidophilic cytoplasm, whereas the distal convoluted tubules (DCTs) had a wider lumen with less acidophilic cytoplasm. Sections from the model group exhibited distorted glomeruli and severe tubular dilation. Multiple hemorrhagic areas were frequently seen among the sections. Furthermore, reference group (NAC) appeared with an amelioration of the glomerular and tubular morphology. Additionally, groups 1, 3, 5, and 6 showed apparent normal renal tissue. In few dark, dense nucleoli were detected in the previous structures. In contrast, groups 4 and 7 showed a moderate affection of glomeruli and tubules with more observable apoptotic cells (arrows) than previously mentioned groups. It was also noticed that groups 2 and 8 displayed severe damage to the glomeruli with severe tubular dilatation (d) which can be seen by during the appearance of more numerous apoptotic cells (arrows) (Figure 10).

##### Supra Renal Cortex

The sham group showed an adrenal gland covered by a connective tissue capsule. Beneath the capsule, the cells of the zona glomerulosa were seen. The next wider zone was the zona fasciculata with a regular arrangement of its cells and the zona reticularis. The model group showed distorted zones with degeneration with frequent dilated arteries. The reference group showed amelioration of all zones, but vacuolations could be seen in most sections. It was noticed that groups 1, 3, 5, and 6 showed apparent normal supra renal tissue, but vacuolations could still be seen. Few dark, dense nucleoli (arrows) could be detected (arrow). Meanwhile, drugs 4 and 7 appeared with moderate thickened capsules (Figure 11). 

##### Renal Tissue

Groups 4, 6, and 8 showed renal tissue nearly approaching the normal renal structure, but a few dark, dense nucleoli could be detected. Groups 4, 6, and 8 exhibited apparent normal supra renal tissue, but vacuolations could still be detected. Few dark dense nucleoli could be detected. Meanwhile, drugs 10 and 7 displayed moderate thicken capsule. More notable apoptotic cells can be seen in them than in the previously mentioned groups. Drugs 5 and 9 showed severe damage to the suprarenal tissue with thickened capsules and interstitial hemorrhaging. More observable apoptotic cells can be seen in them than in the previously mentioned groups (Figure 12).

##### Suprarenal Tissue

Groups 4, 6, and 8 showed apparent normal supra renal tissue, but vacuolations could still be seen. Few dark, dense nucleoli could be noticed. Additionally, groups 7 and 10 showed moderate thickened capsules. More notable apoptotic cells than previously mentioned groups could be seen here. Groups 5 and 9 showed severe damage to the suprarenal tissue. Thickened capsules and hemorrhaging were frequently seen among the sections. More observable apoptotic cells (arrows) can be seen here (Figure 13).

##### Morphometric Study

The renal and suprarenal tissues showed a significant increase in the mean number of apoptotic cells when compared to the control group. However, groups 1, 3, 5, and 6 showed no significant differences when compared to the reference group. Groups 7 and 10 also showed significant differences when compared to the reference group. Groups 5 and 11 showed a highly significant histopathological difference when compared to the reference group (all *p* value ˂ 0.005) (Figure 14).

#### 2.2.6. Molecular Docking Studies Using MOE^®^ Program

The MOE^®^ Dock program was used for performing the molecular docking for compounds under investigation. All tested compounds docked into the binding pocket of the active site of caspase-3 (PDB: 1RHJ) to investigate the docking fitness scores of bioactive conformations and their specificity for the caspase-3 enzyme. The docking reliability was validated using the known X-ray structure of caspase-3 in a complex with PZN. The ligand PZN was extracted from the complex, and the ligand NAC was re-docked to the binding site of caspase-3 (Figure 15). The top-ranked conformation of each compound was selected on the basis of docking score. The docking scores of the studied compounds are shown in Appendix A. 

The molecular docking studies show that most of the tested compounds interacted similarly to the reference NAC, showing a hydrogen bonding interaction with amino acid residues ASP791, GLU789, and LYS259 of the capase-3 enzyme. The tested ligands were found to bind strongly to caspase-3, as inferred by the binding energy values, whose binding scores ranged from −24.58 to −39.77 kcal/mol. Figure 15 shows that the binding mode was similar to the reference NAC for compounds **2b**, which coordinate with caspase-3 through a hydrogen bonding interaction with amino acid residues ASP791 and LYS259. Moreover, both compounds **2a** and **2c** exhibited a hydrogen bonding interaction with the amino acid residue ASP791; however, both of them showed extra binding, two with hydrophobic binding with ARG786 and one with hydrogen bonding with CYS792 amino acid residues, respectively. The superimposition of the active docked poses inside the protein binding pocket for the most active antiapoptotic compounds **2a**, **2c**, and **2e** are shown in Figure 15. Although the docking results for compound **2f** kept hydrogen bonding interactions as well as compound **2b** with LYS259, compounds **2d**, **2e**, **2f**, and **2g** exhibited extra hydrophobic binding with ARG266 and thus can explain the good antiapoptotic activity for these compounds. On the other hand, both of compound **2f** lacked binding with all amino acid residues as the reference; thus, this type of interaction can describe the week antiapoptotic activity for it.

## 3. Conclusions

New thiadiazole derivatives **2a**–**g** were prepared through the cyclization of corresponding hydrazinecarbothioamides **1a**–**g** and were characterized by different spectroscopic techniques. It is worth mentioning that the tested compounds **2c** and **2g** showed more potent activity toward the tested biomarkers in serums and tissues than other thiadiazoles. Histopathological examinations for the renal tissue treated with targeted compounds **1a**–**f** and **2a**–**f** indicated that compounds **2c** and **5g** revealed apparent normal renal cells; however, compound **2d** and **2f** showed dilated blood vessels with more apoptotic cells when compared with NAC. Moreover, caspase-3, 8, and 9 expression was assayed to exhibit that compound **2c** provided lower expression than that of NAC, which showed more potency and selectivity toward caspase-3 than **1c**. Molecular docking studies with the caspase-3 enzyme showed comparable binding scores and similar binding interactions to that of reference NAC. Interestingly, most of the tested compounds showed good binding with the enzyme, especially for compound **2c**. In summary, the conversion of opened hydrazinecarbothioamides into cyclized thiadiazols improves antiapoptotic activity, especially when substituted with 3–5 carbon chains. Compounds **2c** and **2g** are promising antiapoptotic agents that are recommended to be further studied.

## 4. Experimental Section

### 4.1. Chemistry

The IR spectra were recorded by ATR technique (ATR = Attenuated Total Reflection) with an FT device (FT-IR Bruker IFS 88) (Institute of Organic Chemistry, Karlsruhe University, Karlsruhe, Germany). The NMR spectra were measured in DMSO-*d*_6_ on a Bruker AV-400 spectrometer, using 400 MHz for ^1^H and 100 MHz for ^13^C. The chemical shifts are expressed in δ (ppm), versus internal tetramethylsilane (TMS) = 0 for ^1^H and ^13^C, and external liquid ammonia = 0. The description of signals includes: s = singlet, d = doublet, t = triplet, q = quartet, m = multiplet, dd = doublet of doublet, and m = multiplet. Mass spectra were recorded on a FAB (fast atom bombardment) Thermo Finnigan Mat 95 (70 eV). Elemental analyses were carried out at the Microanalytical Center, Cairo University, Egypt. TLC was performed on analytical Merck 9385 silica aluminum sheets (Kieselgel 60) with Pf_254_ indicator. TLCs were viewed at λ_max_ = 254 nm. The synthesized compounds were purified by several recrystallizations from the stated solvent to provide a purity of about 98–99%, confirmed by the elemental analyses and other spectroscopic techniques.

#### 4.1.1. General Method for the Synthesis of Compounds **1a**–**g**

3,4,5-Trimethoxybenzylidenethiocarbonohydrazide (**3**) was prepared according to the published literature [69]. A mixture of **3** (1 mmol) in dry DMF (30 mL) and 1 mmol of substituted isothiocyanates **4a**–**g** was stirred at room temperature for 4–6 h (the reaction was monitored by TLC). After reaction completion, the reaction mixture was poured onto a mixture of crushed ice and water with vigorous stirring. The precipitated solid was filtered, washed with water, dried, and recrystallized from ethanol to afford compounds **1a**–**g**. Substituted-2-(2-(3,4,5-trimethoxybenzylidene)hydrazinecarbonothioyl)hydrazinearbothioamide **1a,b,d**–**g** were prepared according to the literature’s method [70].

(*E*)-*N*-Cyclopropyl-2-(2-(3,4,5-trimethoxybenzylidene)hydrazinecarbonothioyl)hydrazinecarbothioamide (**1c**). Pale yellow crystals (DMF/EtOH), 3.1 g (81%), mp 182–184 °C. IR (KBr) *υ* 3316–3249 (NHs), 3155 (Ar-CH), 2968 (Ali-CH), 1611, 1578 (Ar-C=C), 1530 (NH-def. and C-N str.), 1384, 1125, 932 cm^−1^ (C=S and C-N). ^1^H NMR (DMSO-*d*_6_) *δ_H_* 0.60–0.65 (m, 4H, 2 CH_2_-cyclopropyl), 3.02–3.06 (m, 1H, CH-cyclopropyl), 3.69 (s, 3H, OCH_3_), 3.83 (s, 6H, 2 OCH_3_), 7.14 (s, 2H, Ar-CH), 7.90 (s, 1H, CH=N), 7.97 (s, 1H, NH-cyclopropyl), 9.43 (s, 1H, ^6^NH), 10.13 (s, 1H, ^3^NH), 11.87 ppm (s, 1H, ^4^NH). ^13^C NMR (DMSO-*d_6_*) *δ_C_* 6.28 (CH_2_-cyclopropyl), 27.18 (CH-cyclopropyl), 56.08, 60.05 (OCH_3_), 104.81 (Ar-CH), 129.35, 139.10 (Ar-C), 142.93 (CH=N), 153.07 (Ar-C-O), 179.26, 182.54 ppm (C=S). MS: *m*/*z* 383 (M^+^, 60), 349 (16), 284 (52), 252 (100), 210 (16), 195 (44), 180 (22), 167 (42), 99 (28). *Anal.* Calcd for C_15_H_21_N_5_O_3_S_2_ (383.49): C, 46.98; H, 5.52; N, 18.26; S, 16.72. Found: C, 46.77; H, 5.65; N, 18.39; S, 16.55.

#### 4.1.2. General Method for the Synthesis of Compounds **2a**–**g**

Substituted-2-(2-(3,4,5-trimethoxybenzylidene)hydrazinecarbonothioyl)hydrazine carbothioamides **1a**–**g** (1 mmol) was dissolved in 50 mL absolute ethanol. The mixture was then heated at 60 °C for 2–4 h. Compounds **2a**–**g** precipitated as yellow crystals, were filtered and washed with cold ethanol (20 mL), and recrystallized from the listed solvents.

(*E*)-*N*-Benzyl-5-(2-(3,4,5-trimethoxybenzylidene)hydrazinyl)-1,3,4-thiadiazol-2-amine (**2a**). Yellow crystals (CH_3_CN), 0.359 g (90%), mp 240–242 °C. IR (KBr) *υ* 3310–3280 (NHs), 3141 (Ar-CH), 2928 (Ali-CH), 1626, 1573 (Ar-C=C), 1527 cm^−1^ (NH-def. and C-N str.). ^1^H NMR (DMSO-*d*_6_) *δ_H_* 3.67, 3.83 (s, 9H, 3OCH_3_), 4.43 (d, 2H, CH_2_), 7.13 (m, 2H, Ar-CH), 7.36–7.50 (m, 5H, Ar-CH), 8.20 (s, 1H, CH=N), 9.18 (s, 1H, NH-benzyl), 11.82 ppm (s, 1H, NH). ^13^C NMR (DMSO-*d*_6_) *δ_C_* 45.93 (CH_2_), 55.22, 59.18 (OCH_3_), 103.89, 125.77, 126.28, 128.95 (Ar-CH), 130.58, 137.23, 138.46 (Ar-C), 141.50 (CH=N), 152.23 (Ar-C-O), 159.79 (thiadiazole-C5), 166.66 ppm (thiadiazole-C2). MS: *m*/*z* 399 (M^+^, 100), 307 (9), 195 (32), 149 (86), 99 (17), 91 (38). Anal. Calcd for C_19_H_21_N_5_O_3_S (399.47): C, 57.13; H, 5.30; N, 17.53; S, 8.03. Found: C, 57.05; H, 5.43; N, 17.66; S, 8.09.

(*E*)-*N*-Ethyl-5-(2-(3,4,5-trimethoxybenzylidene)hydrazinecarbonothioyl)hydrazine-carbothioamide (**2b**). Yellow crystals (EtOH), 0.300 g (89%), m.p. 225–227 °C. IR (KBr) *υ* 3306–3254 (NHs), 3140 (Ar-CH), 2930 (Ali-CH), 1617 cm^−1^ (C=N). ^1^H NMR (DMSO-*d*_6_) *δ_H_* 1.07 (t, 3H, CH_3_), 3.48 (q, 2H, CH_2_), 3.68, 3.84 (s, 9H, 3OCH_3_), 7.16 (s, 2H, Ar-CH), 7.95, 7.97 (d, 2H, CH=N and NH-ethyl), 11.55 (s, 1H, NH). ^13^C NMR (DMSO-*d*_6_) *δ_C_* 14.48 (CH_3_), 38.46 (CH_2_), 56.11, 60.05 (OCH_3_), 104.88 (Ar-CH), 129.35, 139.10 (Ar-C), 143.07 (CH=N), 153.07 (Ar-C-O), 160.00 (thiadiazole-C5), 166.64 ppm (thiadiazole-C2). MS: *m*/*z* 337 (M^+^, 100), 308 (15), 293 (30), 200 (55), 195 (10), 154 (40). *Anal.* Calcd for C_14_H_19_N_5_O_3_S (337.40): C, 49.84; H, 5.68; N, 20.76; S, 9.50. Found: C, 49.99; H, 5.56; N, 20.89; S, 9.35.

(*E*)-*N*-Cyclopropyl-5-(2-(3,4,5-trimethoxybenzylidene)hydrazinyl)-1,3,4-thiadiazol-2-amine (**2c**). Yellow crystals (EtOH), 0.317 g (91%), mp 210–212 °C. IR (KBr) *υ* 3305–3250 (NHs), 3142 (Ar-CH), 2933 (Ali-CH), 1618 cm^−1^ (C=N). ^1^H NMR (DMSO-*d*_6_) *δ_H_* 0.73–0.88 (m, 4H, 2 CH_2_-cyclopropyl), 3.07 (m, 1H, CH-cyclopropyl), 3.70, 3.87 (s, 9H, 3 OCH_3_), 7.06 (s, 2H, Ar-CH), 7.96 (s, 1H, CH=N), 8.32 (s, 1H, NH-cyclopropyl), 11.56 ppm (s, 1H, NH). ^13^C NMR (DMSO-*d*_6_) *δ_C_* 6.78 (CH_2_-cyclopropyl), 26.95 (CH-cyclopropyl), 56.05, 60.05 (OCH_3_), 104.77 (Ar-CH), 129.52, 139.01 (Ar-C), 142.04 (CH=N), 153.07 (Ar-C-O), 160.05 (thiadiazole-C5), 166.64 ppm (thiadiazole-C2). MS: *m*/*z* 349 (M^+^, 100), 308 (21), 276 (15), 256 (9), 195 (40), 154 (47). *Anal.* Calcd for C_15_H_19_N_5_O_3_S (349.41): C, 51.56; H, 5.48; N, 20.04; S, 9.18. Found: C, 51.77; H, 5.64; N, 19.89; S, 9.06.

(*E*)-*N*-Phenyl-5-(2-(3,4,5-trimethoxybenzylidene)hydrazinyl)-1,3,4-thiadiazol-2-amine (**2d**). Yellow crystals (EtOH), 0.362 g (94%), mp 220–222 °C. IR (KBr) *υ* 3312–3270 (NHs), 2996 (Ar-CH), 1621 cm^−1^ (C=N). ^1^H NMR (DMSO-*d*_6_) *δ_H_* 3.69, 3.83, (s, 9H, 3 OCH_3_), 6.91–6.95 (s, 3H, Ar-CH), 7.28–7.32 (m, 2H, Ar-CH), 7.54–7.56 (m, 2H, Ar-CH), 7.95 (s, 1H, CH=N), 9.83 (s, 1H, NH-phenyl), 11.93 ppm (s, 1H, NH). ^13^C NMR (DMSO-*d_6_*) *δ_C_* 56.21, 60.57 (OCH_3_), 103.85, 117.19, 121.38, 129.41 (Ar-CH), 130.54, 138.96, 141.59 (Ar-C), 142.75 (CH=N), 153.61 (Ar-C-O), 163.22 (thiadiazole-C5), 165.12 ppm (thiadiazole-C2). MS: *m*/*z* 385 (M^+^, 28), 192 (40), 176 (16), 165 (32), 130 (10), 116 (26), 91 (100), 77 (36). *Anal.* Calcd for C_18_H_19_N_5_O_3_S (385.44): C, 56.09; H, 4.97; N, 18.17; S, 8.32. Found: C, 55.97; H, 4.92; N, 18.29; S, 8.41. 

(*E*)-*N*-Cyclohexyl-5-(2-(3,4,5-trimethoxybenzylidene)hydrazinyl)-1,3,4-thiadiazol-2-amine (**2e**). Yellow crystals (CH_3_CN), 0.363 g (93%), mp 235–237 °C. IR (KBr) *υ* 3390–3280 (NHs), 3087 (Ar-CH), 1625 cm^−1^ (C=N). ^1^H NMR (DMSO-*d*_6_) *δ_H_* 1.67 (m, 4H, 2 cyclohexyl-CH_2_), 1.94 (m, 6H, 3 cyclohexyl-CH_2_), 2.73 (m, 1H, cyclohexyl-CH), 3.68, 3.87 (s, 9H, 3 OCH_3_), 7.12 (s, 2H, Ar-CH), 7.90 (s, 1H, CH=N), 7.93 (s, 1H, NH-hexyl), 11.90 ppm (s, 1H, NH). ^13^C NMR (DMSO-*d*_6_) *δ_C_* 25.31, 32.18, 35.74 (cyclohexyl-CH_2_), 52.81 (cyclohexyl-CH), 56.08, 60.09 (OCH_3_), 104.83 (Ar-CH), 130.30, 138.25, 139.13 (Ar-C), 143.03 (CH=N), 153.15 (Ar-C-O), 160.25 (thiadiazole-C5), 165.13 ppm (thiadiazole-C2). MS: *m*/*z* 391 (M^+^, 20), 307 (38), 281 (16), 168 (5), 156 (100). *Anal.* Calcd for C_18_H_25_N_5_O_3_S (391.49): C, 55.22; H, 6.44; N, 17.89; S, 8.19. Found: C, 55.35; H, 6.32; N, 18.01; S, 8.11. 

(*E*)-*N*-Allyl-5-(2-(3,4,5-trimethoxybenzylidene)hydrazinyl)-1,3,4-thiadiazol-2-amine (**2f**). Yellow crystals (CHCl_3_/EtOH), 0.310 g (89%), mp 209–211 °C. IR (KBr) υ =3309–3252 (NHs), 3173 (Ar-CH), 2933 (Ali-CH), 1621 (C=N) cm^−1^. ^1^H NMR (DMSO-*d*_6_) *δ_H_* 3.69, 3.83 (s, 9H, 3 OCH_3_), 4.21–4.24 (m, 2H, CH_2_N), 5.11–5.19 (m, 2H, allyl-CH_2_), 5.91–5.99 (m, 1H, allyl-CH), 7.07 (s, 2H, Ar-CH), 7.99 (s, 1H, CH=N), 8.62 (s, 1H, NH-allyl), 11.57 ppm (s, 1H, NH). ^13^C NMR (DMSO-*d*_6_) *δ_C_* 45.71 (CH_2_N), 56.10, 60.27 (OCH_3_), 104.71 (Ar-CH), 115.40 (allyl-CH_2_), 129.21 (Ar-C), 135.13 (allyl-CH=), 139.00, 140.12 (Ar-C), 142.17 (CH=N), 153.11 (Ar-C-O), 161.22 (thiadiazole-C5), 164.41 ppm (thiadiazole-C2). MS: *m*/*z* 349 (M^+^, 32), 308 (58), 276 (5), 195 (45), 155 (100). *Anal.* Calcd for C_15_H_19_N_5_O_3_S (349.41): C, 51.56; H, 5.48; N, 20.04; S, 9.18. Found: C, 51.65; H, 5.39; N, 20.18; S, 9.07.

(*E*)-*N*-Pentyl-5-(2-(3,4,5-trimethoxybenzylidene)hydrazinyl)-1,3,4-thiadiazol-2-amine (**2g**). Yellow crystals (CH_3_OH), 0.348 g (92%), mp 214–216 °C. IR (KBr) *υ* 305–3272 (NH), 3146 (Ar-CH), 2929 (Ali-CH), 1613 cm^−1^ (C=N). ^1^H NMR (DMSO-*d*_6_) *δ_H_* 0.88 (t, 3H, CH_3_), 1.33 (m, 4H, 2 CH_2_), 1.60 (m, 2H, CH_2_), 3.56 (m, 2H, CH_2_), 3.70, 3.86 (s, 9H, 3 OCH_3_), 7.04 (s, 2H, Ar-CH), 7.99 (s, 1H, CH=N), 8.50 (s, 1H, NH-pentyl) 11.50 ppm (s, 1H, NH). ^13^C NMR (DMSO-*d*_6_) *δ_C_* 13.91 (CH_3_), 22.10, 28.36, 28.39, 43.40 (CH_2_-pentyl) 56.10, 60.01 (OCH_3_), 104.85 (Ar-CH), 129.99, 139.97, 140.01 (Ar-C), 143.01 (CH=N), 153.04 (Ar-C-O), 161.80 (thiadiazole-C5), 165.00 ppm (thiadiazole-C2). MS: *m/z* 379 (M^+^, 100), 336 (46), 307 (7), 286 (5), 204(8), 195 (24), 154 (9). *Anal.* Calcd for C_17_H_25_N_5_O_3_S (379.48): C, 53.81; H, 6.64; N, 18.46; S, 8.45. Found: C, 53.77; H, 6.70; N, 18.60; S, 8.57.

### 4.2. Single Crystal X-ray Structure Determination of ***2d***

Single crystals were obtained by recrystallization from acetonitrile. The single crystal X-ray diffraction study was carried out on a Bruker D8 VENTURE diffractometer with a PhotonII CPAD detector at 298 K using Cu *K*α radiation (λ = 1.54178 Å). Dual space methods (SHELXT) [71] were used for the structure solution, and refinement was carried out using SHELXL [72] (full-matrix least-squares on F^2^). Hydrogen atoms were localized by difference electron density determination and refined using a riding model (H(N) free). A semi-empirical absorption correction was applied.

**2d**: C_18_H_19_N_5_O_3_S, *M* = 385.44 g mol^−1^, yellow crystal, size 0.32 × 0.08 × 0.02 mm, monoclinic space group P2_1_/n (no.14), *a* = 17.8658 (3) Å, *b* = 5.3161(2) Å, *c* = 19.6971 (3) Å, *β* = 96.408 (1) Å, *V* = 1860.99 (5) Å^3^, *Z =* 4, *D_calcd_* = 1.376 mg m^−3^, *F*(000) = 808, *μ* = 1.80 mm^−1^, *T* = 298 K, 15205 measured reflection (2θ_max_ = 144.8°), 3683 independent [*R_int_ =* 0.031], 253 parameters, 2 restraint, *R_1_* [for 3428 *I* > 2σ(1)] = 0.032, *wR^2^* (for all data) = 0.092, *S* = 1.07, largest diff. peak and hole = 0.22 eÅ^−3^/−0.23 eÅ^−3^.

CCDC 2056821 (**2d**) contains the supplementary crystallographic data for this paper. These data can be obtained free of charge from The Cambridge Crystallographic Data Centre via www.ccdc.cam.ac.uk/data_request/cif (deposited 18 January 2021).

### 4.3. Biological Evaluation

#### 4.3.1. Materials and Methods of Biomarkers (Creatinine, Urea, Ep, MDA, TAC, NOx)

The concentration of EP was determined in serum using the spectrophotofluorometric method (Shimadzu RF-5000). Transmitter’s oxidations were conducted by adding 0.1N iodine, followed by stopping the oxidation by alkaline sulfite addition to produce a certain fluorescence. The induced fluorescence measured at a specific emission wavelength after excitation at another specific wavelength differs according to the type of transmitter [73] (see Appendix A). Determination of serum urea was assayed by using an enzymatic colorimetric urea kit.

Principle of measuring urea and ammonia in the chosen sample*:* The Berthelot reactionhas long been used for the measurement of urea and ammonia [74]. This method is a modification of the Berthelot reaction, in which phenol is replaced by salicylic acid. Urea in the sample was hydrolyzed by the enzyme urease to yield ammonia and carbon dioxide. The ammonium ions then reacted with a mixture of salicylate, sodium nitroprusside, and sodium hypochloride to yield a green indophenol. (See Appendix A).

Assessment of Total Nitrite in Kidney: The stable oxidation end products of NO, nitrite (NO_2_¯), and nitrate (NO_3_¯) were used in vitro and in vivo as indicators of NO production. Thus, NO_2_¯ and NO_3_¯ levels were estimated as an index of NO production. Total nitrite (nitrite + nitrate) was measured after the reduction of nitrate to nitrite by copperized cadmium granules (Cd) in a glycine buffer at pH 9.7 (2.5 to 3.0 g of Cd granules for a 4 mL reaction mixture). Quantitation of NO_2_¯ was based on the Griess reaction, in which a chromophore with a strong absorbance at 540 nm is formed by the reaction of nitrite with a mixture of naphthylethylenediamine and sulphanilamide [75] (See Appendix A).

#### 4.3.2. Assay of Caspase-3, 8, and 9 Inhibition

The Caspase-3 Inhibitor Drug Screening Kit (Catalog #JM-K153-100; 15 B Constitution Way, Woburn, 01801-MA, USA) provides an effective means for screening caspase inhibitors using fluorometric methods. The assay utilizes the synthetic peptide substrate DEVD-AFC (AFC, 7-amino-4-trifluoromethylcoumarin). Active caspase-3 cleaves the synthetic substrate to release free AFC, which can then be quantified by fluorometry. Compounds to be screened can be directly added to the reaction, and the level of inhibition of caspase-3 activity can be determined by comparison of the fluorescence intensity in samples with and without the testing inhibitors (See Appendix A).

#### 4.3.3. Assay of Cytochrome *C*

Cells were collected from American Type Culture Collection to be grown in SR containing 10% fetal bovine serum at 37 °C, stimulated with the compounds to be tested for cytochrome *C* using Cytochrome *C* Human ELISA Kit (ab119521 –Cytochrome *C* Human ELISA Kit, Vaccera institute, Cairo, Egypt) [76] (See Appendix A).

#### 4.3.4. Histopathological Investigation

Multiple testicular specimens were stained by hematoxylin-eosin (ab245880, Abcam, Cambridge, MA, USA) using Cosentino’s score [76] for the semi-quantitation of pathological changes in different seminiferous tubules. By using Johnson’s scoring system [76], the effect of ischemia on spermatogenesis can be studied (See Appendix A).

#### 4.3.5. Molecular Docking Study

The docking simulation study was carried out using the Molecular Operating Environment (MOE^®^) version 2014.09 (Chemical Computing Group Inc., Montreal, QC, Canada). The computational software operated under “Windows XP”, installed on an Intel Pentium IV PC with a 1.6 GHz processor and 512 MB memory. (See Appendix A).

## Data Availability

All pertinent data have been supplied in the Supporting Information that accompanies this article.

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
