# Peer review of "Synthesis, Characterization, and In Vivo Study of Some Novel 3,4,5-Trimethoxybenzylidene-hydrazinecarbothioamides and Thiadiazoles as Anti-Apoptotic Caspase-3 Inhibitors"

_molecules, 2022, doi:10.3390/molecules27072266_

Round 1

Reviewer 1 Report

The authors describe the potential use of 3,4,5-trimethoxybenzylidene-hydrazinecarbothioamides and thiadiazoles as anti-apoptotic caspase-3 inhibitors. They describe the synthesis and characterization of seven acyclic 3,4,5-trimethoxybenzylidene-hydrazinecarbothioamides and seven cyclic thiadiazoles. They describe the useful SAR of these analogues and the experiments show its potential anti-apoptotic activity for the treatment of acute renal ischemia. These findings are novel and would be of interest to the scientific community. However, there are some issues that need to be addressed hence I recommend the publication be accepted to Molecules with minor corrections to the below has been made.
Line 28- Consider removing “confirmation of the structure via spectroscopic tools including NMR, IR and mass spectra together with elemental analysis”. It is an abstract and should be concise.
Line 49- What are “these processes”. Explain them, as it is the first sentence of your introduction.
Line 109- You have selected 1a-g and 2a-g as your target compounds. How did you come to this scaffold? Figure 1 shows the common drugs used for this purpose but it does not show how you came to make 1a-g and 2a-g. You need to need explain how you selected this scaffold (HTS? Initial lead?).
Line 129- Three methoxy group should have 3 peaks, obviously 3.83 ppm has 2 peaks overlapping, so mention it is overlapped.
Line 136- Please state yields of each step in the Scheme.
Line 159- Structure of 2d was confirmed by X-ray crystallography which is in the E-form. How about the other analogues? Can we assume they all adopt the E-form? Can we look at coupling constants of the double bond to determine whether they are E or Z isomers?
Line 244- Bold ‘1c’
Line 290- Figure 9 has typo- ‘ant’ correct to ‘anti’, ‘decreased’
Line 414- the sentence is not complete, missing text?
Line 450- The compounds that were synthesized, did they get HPLC purity checks? How were the compounds checked for purity levels?
Supporting info- The 1H NMR spectra are missing integrals. Please insert the integrals.
References- please check the formatting of journal name, years, issues etc and change them to format required by Molecules. There are some journals with abbreviated journals while some are not.

Author Response

1

Line 28- Consider removing “confirmation of the structure via spectroscopic tools including NMR, IR and mass spectra together with elemental analysis”. It is an abstract and should be concise.

The sentence “ … including NMR, IR and mass spectra together with…” is omitted and the word “and” is added

2

Line 49- What are “these processes”. Explain them, as it is the first sentence of your introduction

3

Line 109- You have selected 1a-g and 2a-g as your target compounds. How did you come to this scaffold?

Figure 1 shows the common drugs used for this purpose but it does not show how you came to make 1a-g and 2a-g. You need to need explain how you selected this scaffold (HTS? Initial lead?).

As mentioned in the paragraph of (line 90-100 )

Thiadiazole scaffold revealed a protective action against ischemia via inhibition of NF-κB, apoptosis as well as other heterocycles, such as triazolopyrimidine derivative II (Figure 1) to be tested in a kidney ischemia–reperfusion modes. We hyperdize this findings and decide to synthesize new thiadiazole analogs (2a-g) and study their anti-apoptotic activity in comparison to their precursor including a mechanistic study (1a-g).

Modification to figure 1 is done .

4

Line 129- Three methoxy group should have 3 peaks, obviously 3.83 ppm has 2 peaks overlapping, so mention it is overlapped.

The sentence  “….as two methoxy protons sets are overlapped…” is added

5

Line 136- Please state yields of each step in the Scheme

Yields are added

6

Line 159- Structure of 2d was confirmed by X-ray crystallography which is in the E-form. How about the other analogues? Can we assume they all adopt the E-form? Can we look at coupling constants of the double bond to determine whether they are E or Z isomers?

The structure of the compounds were corrected (please have a look at Scheme 1). Most accurate structures of 2a-g would have similar structure as in 2d. 15N-H and long-range couplings are required for all analogues, which were not done.   

7

Line 244- Bold ‘1c’

“1c” is bolded

8

Line 290- Figure 9 has typo- ‘ant’ correct to ‘anti’, ‘decreased’

The words ‘ant-apoptotic’ is corrected to ‘anti-apoptotic’ and ‘deceased’ to ‘decreased

9

Line 414- the sentence is not complete, missing text?

Sentence is corrected into “Hence, it could explain the good antiapoptotic activity for these compounds. On the other hand, both compound 2f lacked binding with all amino acid residues as the reference, thus, this type of interactions could describe the weak anti-apoptotic activity for it.”

10

Line 450- The compounds that were synthesized, did they get HPLC puritychecks? How were the compounds checked for purity levels?

We did not get HPLC purity check however, The synthesized compounds were purified by several recrystallization from the stated solvent to provide purity about 98-99%. That was confirmed by the elemental analyses and other spectroscopic techniques. Also, This sentence is added.

Note:

NCI (National Cancer Institutes) accepts compound samples with 95% purity to carry out biological anticancer investigation which means our purity percent does not conflict with biological work.

11

Supporting info- The 1H NMR spectra are missing integrals. Please insert the integrals.

Integrals are added to The 1H NMR spectra

12

References- please check the formatting of journal name, years, issues etc and change them to format required by Molecules. There are some journals with abbreviated journals while some are not.

References are checked according to Molecule template and abbreviation are corrected

Reviewer 2 Report

The manuscript entitled “Synthesis, characterization and in vivo study of some novel 3,4,5-trimethoxybenzylidene-hydrazinecarbothioamides and thiadiazoles as anti-apoptotic caspase-3 inhibitors” by Sara M. Mostafa et al. described an approach for the synthesis of a series of new thiadiazole analogues based on (E)-N-(substituted)-2-(2-(3,4,5-trimethoxybenzylidene)hydrazinecarbonothioyl)hydrazinecarbothioamides and study their anti-apoptotic caspase-3 inhibition activity. The manuscript may be of general interest to the researchers of this field, but the author should consider and incorporate in the present form of the manuscript a few concerns that need to be corrected.

  1. The abstract should be shortened (max 200 words, see rules of journal).
  2. The authors should check image of E-form for compounds 1a-g in the Scheme 1.
  3. The reference [72] should be corrected.

Author Response

1

The abstract should be shortened (max 200 words, see rules of journal).

2

The authors should check image of E-form for compounds 1a-g in the Scheme 1.

The most probably expected structure of compounds 1a-g was changed. The two sulfur atoms of the two thiones must be in the same direction in order to facilitate the cyclization process. Moreover the structure of compounds structures of compounds 2a-g were also corrected to be accommodated with the X-ray structure analysis o compound 2d

3

The reference [72] should be corrected

The reference is corrected